# CSKG: The CommonSense Knowledge Graph

Filip Ilievski, Pedro Szekely, and Bin Zhang

Information Sciences Institute, University of Southern California
{ilievski,pszekely,binzhang}@isi.edu

**Abstract.** Sources of commonsense knowledge support applications in natural language understanding, computer vision, and knowledge graphs. Given their complementarity, their integration is desired. Yet, their different foci, modeling approaches, and sparse overlap make integration difficult. In this paper, we consolidate commonsense knowledge by following five principles, which we apply to combine seven key sources into a first integrated CommonSense Knowledge Graph (CSKG). We analyze CSKG and its various text and graph embeddings, showing that CSKG is well-connected and that its embeddings provide a useful entry point to the graph. We demonstrate how CSKG can provide evidence for generalizable downstream reasoning and for pre-training of language models. CSKG and all its embeddings are made publicly available to support further research on commonsense knowledge integration and reasoning.

**Resource type**: Knowledge graph
**License**: CC BY-SA 4.0
**DOI**: https://doi.org/10.5281/zenodo.4331372
**Repository**: https://github.com/usc-isi-i2/cskg

**Keywords:** commonsense knowledge · knowledge graph · embeddings

## 1 Introduction

Recent commonsense reasoning benchmarks [27,3] and neural advancements [17,16] shed a new light on the longstanding task of capturing, representing, and reasoning over commonsense knowledge. While state-of-the-art language models [8,17] capture linguistic patterns that allow them to perform well on commonsense reasoning tasks after fine-tuning, their robustness and explainability could benefit from integration with structured knowledge, as shown by KagNet [16] and HyKAS [18]. Let us consider an example task question from the SWAG dataset [38],[1] which describes a woman that takes a sit at the piano:

```
Q: On stage, a woman takes a seat at the piano. She:
1. sits on a bench as her sister plays with the doll.
2. smiles with someone as the music plays.
3. is in the crowd, watching the dancers.
-> 4. nervously sets her fingers on the keys.
```

---

[1] The multiple-choice task of choosing an intuitive follow-up scene is customary called question answering [19,38], despite the absence of a formal question.

Answering this question requires knowledge that humans possess and apply, but machines cannot distill directly in communication. Luckily, graphs of (commonsense) knowledge contain such knowledge. ConceptNet's [29] triples state that pianos have keys and are used to perform music, which supports the correct option and discourages answer 2. WordNet [21] states specifically, though in natural language, that pianos are played by pressing keys. According to an image description in Visual Genome, a person could play piano while sitting and having their hands on the keyboard. In natural language, ATOMIC [26] indicates that before a person plays piano, they need to sit at it, be on stage, and reach for the keys. ATOMIC also lists strong feelings associated with playing piano. FrameNet's [1] frame of a performance contains two separate roles for the performer and the audience, meaning that these two are distinct entities, which can be seen as evidence against answer 3.

While these sources clearly provide complementary knowledge that can help commonsense reasoning, their different foci, representation formats, and sparse overlap makes integration difficult. Taxonomies, like WordNet , organize conceptual knowledge into a hierarchy of classes. An independent ontology, coupled with rich instance-level knowledge, is provided by Wikidata [34], a structured counterpart to Wikipedia. FrameNet, on the other hand, defines an orthogonal structure of frames and roles; each of which can be filled with a WordNet/Wikidata class or instance. Sources like ConceptNet or WebChild [31], provide more 'episodic' commonsense knowledge, whereas ATOMIC captures pre- and post-situations for an event. Image description datasets, like Visual Genome [14], contain visual commonsense knowledge. While links between these sources exist (mostly through WordNet synsets), the majority of their nodes and edges are disjoint.

In this paper, we propose an approach for integrating these (and more sources) into a single Common Sense Knowledge Graph (CSKG). We suvey existing sources of commonsense knowledge to understand their particularities and we summarize the key challenges on the road to their integration (section 2). Next, we devise five principles and a representation model for a consolidated CSKG (section 3). We apply our approach to build the first version of CSKG, by combining seven complementary, yet disjoint, sources. We compute several graph and text embeddings to facilitate reasoning over the graph. In section 4, we analyze the content of the graph and the generated embeddings. We provide insights into the utility of CSKG for downstream reasoning on commonsense Question Answering (QA) tasks in section 5. In section 6 we reflect on the learned lessons and list the next steps for CSKG. We conclude in section 7.

## 2   Problem statement

### 2.1   Sources of Common Sense Knowledge

Table 1 summarizes the content, creation method, size, external mappings, and example resources for representative public commonsense sources: ConceptNet [29], WebChild [31], ATOMIC [26], Wikidata [34], WordNet [21], Roget [13], VerbNet [28], FrameNet [1], Visual Genome [14], and ImageNet [7]. Primarily, we

**Table 1.** Survey of existing sources of commonsense knowledge.

|  | describes | creation | size | mappings | examples |
|---|---|---|---|---|---|
| **Concept Net** | everyday objects, actions, states, relations (multilingual) | crowd-sourcing | 36 relations, 8M nodes, 21M edges | WordNet, DBpedia, OpenCyc, Wiktionary | `/c/en/piano` `/c/en/piano/n` `/c/en/piano/n/wn` `/r/relatedTo` |
| **Web Child** | everyday objects, actions, states, relations | curated automatic extraction | 4 relation groups, 2M nodes, 18M edges | WordNet | `hasTaste` `fasterThan` |
| **ATOMIC** | event pre/post-conditions | crowd-sourcing | 9 relations, 300k nodes, 877k edges | ConceptNet, Cyc | `wanted-to` `impressed` |
| **Wikidata** | instances, concepts, relations | crowd-sourcing | 1.2k relations, 75M objects, 900M edges | various | `wd:Q1234 wdt:P31` |
| **WordNet** | words, concepts, relations | manual | 10 relations, 155k words, 176k synsets |  | `dog.n.01` `hypernymy` |
| **Roget** | words, relations | manual | 2 relations, 72k words, 1.4M edges |  | `truncate` `antonym` |
| **VerbNet** | verbs, relations | manual | 273 top classes 23 roles, 5.3k senses | FrameNet, WordNet | `perform-v` `performance-26.7-1` |
| **FrameNet** | frames, roles, relations | manual | 1.9k edges, 1.2k frames, 12k roles, 13k lexical units |  | `Activity` `Change_of_leadership` `New_leader` |
| **Visual Genome** | image objects, relations, attributes | crowd-sourcing | 42k relations, 3.8M nodes, 2.3M edges, 2.8M attributes | WordNet | `fire hydrant` `white dog` |
| **ImageNet** | image objects | crowd-sourcing | 14M images, 22k synsets | WordNet | `dog.n.01` |

observe that the commonsense knowledge is spread over a number of sources with different focus: commonsense knowledge graphs (e.g., ConceptNet), general-domain knowledge graphs (e.g., Wikidata), lexical resources (e.g., WordNet, FrameNet), taxonomies (e.g., Wikidata, WordNet), and visual datasets (e.g., Visual Genome) [11]. Therefore, these sources together cover a rich spectrum of knowledge, ranging from everyday knowledge, through event-centric knowledge and taxonomies, to visual knowledge. While the taxonomies have been created manually by experts, most of the commonsense and visual sources have been created by crowdsourcing or curated automatic extraction. Commonsense and common knowledge graphs (KGs) tend to be relatively large, with millions of nodes and edges; whereas the taxonomies and the lexical sources are notably smaller. Despite the diverse nature of these sources, we note that many contain mappings to WordNet, as well as a number of other sources. These mappings might be incomplete, e.g., only a small portion of ATOMIC can be mapped to ConceptNet. Nevertheless, these high-quality mappings provide an opening for consolidation of commonsense knowledge, a goal we pursue in this paper.

## 2.2   Challenges

Combining these sources in a single KG faces three key challenges:

1. The sources follow **different knowledge modeling approaches**. One such difference concerns the relation set: there are very few relations in ConceptNet and WordNet, but (tens of) thousands of them in Wikidata and Visual Genome. Consolidation requires a global decision on how to model the relations. The granularity of knowledge is another factor of variance. While regular RDF triples fit some sources (e.g., ConceptNet), representing entire frames (e.g., in FrameNet), event conditions (e.g., in ATOMIC), or compositional image data (e.g., Visual Genome) might benefit from a more open format. An ideal representation would support the entire granularity spectrum.

2. As a number of these sources have been created to support natural language applications, they often contain **imprecise descriptions**. Natural language phrases are often the main node types in the provided knowledge sources, which provides the benefit of easier access for natural language algorithms, but it introduces ambiguity which might be undesired from a formal semantics perspective. An ideal representation would harmonize various phrasings of a concept, while retaining easy and efficient linguistic access to these concepts via their labels.

3. Although these sources contain links to existing ones, we observe **sparse overlap**. As these external links are typically to WordNet, and vary in terms of their version (3.0 or 3.1) or target (lemma or synset), the sources are still disjoint and establishing (identity) connections is difficult. Bridging these gaps, through optimally leveraging existing links, or extending them with additional ones automatically, is a modeling and integration challenge.

## 2.3   Prior consolidation efforts

Prior efforts that combine pairs or small sets of (mostly lexical) commonsense sources exist. A unidirectional manual mapping from VerbNet classes to WordNet and FrameNet is provided by the Unified Verb Index [33]. The Predicate Matrix [6] has a full automatic mapping between lexical resources, including FrameNet, WordNet, and VerbNet. PreMOn [5] formalizes these in RDF. In [20], the authors produce partial mappings between WordNet and Wikipedia/DBpedia. Zareian et al. [37] combine edges from Visual Genome, WordNet, and ConceptNet to improve scene graph generation from an image. None of these efforts aspires to build a consolidated KG of commonsense knowledge.

Most similar to our effort, BabelNet [22] integrates many sources, covers a wide range of 284 languages, and primarily focuses on lexical and general-purpose resources, like WordNet, VerbNet, and Wiktionary. While we share the goal of integrating valuable sources for downstream reasoning, and some of these sources (e.g., WordNet) overlap with BabelNet, our ambition is to support commonsense reasoning applications. For this reason, we focus on commonsense knowledge graphs, like ConceptNet and ATOMIC, or even visual sources, like Visual Genome, none of which are found in BabelNet.

## 3   The Common Sense Knowledge Graph

### 3.1   Principles

Question answering and natural language inference tasks require knowledge from heterogeneous sources (section 2). To enable their joint usage, the sources need to be harmonized in a way that will allow straightforward access by linguistic tools [18,16], easy splitting into arbitrary subsets, and computation of common operations, like (graph and word) embeddings or KG paths. For this purpose, we devise five principles for consolidatation of sources into a single commonsense KG (CSKG), driven by pragmatic goals of simplicity, modularity, and utility:

**P1. Embrace heterogeneity of nodes** One should preserve the natural node diversity inherent to the variety of sources considered, which entails blurring the distinction between objects (such as those in Visual Genome or Wikidata), classes (such as those in WordNet or ConceptNet), words (in Roget), actions (in ATOMIC or ConceptNet), frames (in FrameNet), and states (as in ATOMIC). It also allows formal nodes, describing unique objects, to co-exist with fuzzy nodes describing ambiguous lexical expressions.

**P2. Reuse edge types across resources** To support reasoning algorithms like KagNet [16], the set of edge types should be kept to minimum and reused across resources wherever possible. For instance, the ConceptNet edge type `/r/LocatedNear` could be reused to express spatial proximity in Visual Genome.

**P3. Leverage external links** The individual graphs are mostly disjoint according to their formal knowledge. However, high-quality links may exist or may be easily inferred, in order to connect these KGs and enable path finding. For instance, while ConceptNet and Visual Genome do not have direct connections, they can be partially aligned, as both have links to WordNet synsets.

**P4. Generate high-quality probabilistic links** Inclusion of additional probabilistic links, either with off-the-shelf link prediction algorithms or with specialized algorithms (e.g., see section 3.3), would improve the connectedness of CSKG and help path finding algorithms reason over it. Given the heterogeneity of nodes (cf. P1), a 'one-method-fits-all' node resolution might not be suitable.

**P5. Enable access to labels** The CSKG format should support easy and efficient natural language access. Labels and aliases associated with KG nodes provide application-friendly and human-readable access to the CSKG, and can help us unify descriptions of the same/similar concept across sources.

### 3.2   Representation

We model CSKG as a **hyper-relational graph**, describing edges in a tabular KGTK [10] format. We opted for this representation rather than the traditional RDF/OWL2 because it allows us to fulfill our goals (of simplicity and utility) and follow our principles more directly, without compromising on the format. For instance, natural language access (principle P5) to RDF/OWL2 nodes requires graph traversal over its `rdfs:label` relations. Including both reliable and probabilistic nodes (P3 and P4) would require a mechanism to easily indicate edge

weights, which in RDF/OWL2 entails inclusion of blank nodes, and a number of additional edges. Moreover, the simplicity of our tabular format allows us to use standard off-the-shelf functionalities and mature tooling, like the `pandas`[2] and `graph-tool`[3] libraries in Python, or graph embedding tools like [15], which have been conveniently wrapped by the KGTK [10] toolkit.[4]

The edges in CSKG are described by ten columns. Following KGTK, the primary information about an edge consists of its `id`, `node1`, `relation`, and `node2`. Next, we include four "lifted" edge columns, using KGTK's abbreviated way of representing triples about the primary elements, such as `node1;label` or `relation;label` (label of `node1` and of `relation`). Each edge is completed by two qualifiers: `source`, which specifies the source(s) of the edge (e.g., "CN" for ConceptNet), and `sentence`, containing the linguistic lexicalization of a triple, if given by the original source. Auxiliary KGTK files can be added to describe additional knowledge about some edges, such as their weight, through the corresponding edge `ids`. We provide further documentation at: `https://cskg.readthedocs.io/`.

### 3.3   Consolidation

Currently, CSKG integrates seven sources, selected based on their popularity in existing QA work: a commonsense knowledge graph ConceptNet, a visual commonsense source Visual Genome, a procedural source ATOMIC, a general-domain source Wikidata, and three lexical sources, WordNet, Roget, and FrameNet. Here, we briefly present our design decisions per source, the mappings that facilitate their integration, and further refinements on CSKG.

**3.3.1   Individual sources** We keep the original edges of **ConceptNet** 5.7 expressed with 47 relations in total. We also include the entire **ATOMIC** KG, preserving the original nodes and its nine relations. To enhance lexical matching between ATOMIC and other sources, we add normalized labels of its nodes, e.g., adding a second label "accepts invitation" to the original one "personX accepts personY's invitation". We import four node types from **FrameNet**: frames, frame elements (FEs), lexical units (LUs), and semantic types (STs), and we reuse 5 categories of FrameNet edges: frame-frame (13 edge types), frame-FE (1 edge type), frame-LU (1 edge type), FE-ST (1 edge type), and ST-ST (3 edge types). Following principle P2 on edge type reuse, we map these 19 edge types to 9 relations in ConceptNet, e.g., `is_causative_of` is converted to `/r/Causes`. **Roget** We include all synonyms and antonyms between words in Roget, by reusing the ConceptNet relations `/r/Synonym` and `/r/Antonym` (P2). We represent **Visual Genome** as a KG, by representing its image objects as WordNet synsets (e.g., `wn:shoe.n.01`). We express relationships between objects via ConceptNet's `/r/LocatedNear` edge type. Object attributes are represented by

---

[2]`https://pandas.pydata.org/`

[3]`https://graph-tool.skewed.de/`

[4]CSKG can be transformed to RDF with `kgtk generate-wikidata-triples`.

different edge types, conditioned on their part-of-speech: we reuse ConceptNet's `/r/CapableOf` for verbs, while we introduce a new relation `mw:MayHaveProperty` for adjective attributes. We include the *Wikidata-CS* subset of **Wikidata**, extracted in [12]. Its 101k statements have been manually mapped to 15 ConceptNet relations. We include four relations from **WordNet** v3.0 by mapping them to three ConceptNet relations: hypernymy (using `/r/IsA`), part and member holonymy (through `/r/PartOf`), and substance meronymy (with `/r/MadeOf`).

**3.3.2   Mappings**  We perform node resolution by applying existing identity mappings (P3) and generating probabilistic mappings automatically (P4). We introduce a dedicated relation, `mw:SameAs`, to indicate identity between two nodes.
**WordNet-WordNet** The WordNet v3.1 identifiers in ConceptNet and the WordNet v3.0 synsets from Visual Genome are aligned by leveraging ILI: the WordNet InterLingual Index,[5] which generates 117,097 `mw:SameAs` mappings.
**WordNet-Wikidata** We generate links between WordNet synsets and Wikidata nodes as follows. For each synset, we retrieve 50 candidate nodes from a customized index of Wikidata. Then, we compute sentence embeddings of the descriptions of the synset and each of the Wikidata candidates by using a pretrained XLNet model [36]. We create a `mw:SameAs` edge between the synset and the Wikidata candidate with highest cosine similarity of their embeddings. Each mapping is validated by one student. In total, 17 students took part in this validation. Out of the 112k edges produced by the algorithm, the manual validation marked 57,145 as correct. We keep these in CSKG and discard the rest.
**FrameNet-ConceptNet** We link FrameNet nodes to ConceptNet in two ways. FrameNet LUs are mapped to ConceptNet nodes through the Predicate Matrix [6] with $3,016$ `mw:SameAs` edges. Then, we use 200k hand-labeled sentences from the FrameNet corpus, each annotated with a target frame, a set of FEs, and their associated words. We treat these words as LUs of the corresponding FE, and ground them to ConceptNet with the rule-based method of [16].
**Lexical matching** We establish 74,259 `mw:SameAs` links between nodes in ATOMIC, ConceptNet, and Roget by exact lexical match of their labels. We restrict this matching to lexical nodes (e.g., `/c/en/cat` and not `/c/en/cat/n/wn/animal`).

**3.3.3   Refinement**  We consolidate the seven sources and their interlinks as follows. After transforming them to the representation described in the past two sections, we concatenate them in a single graph. We deduplicate this graph and append all mappings, resulting in `CSKG*`. Finally, we apply the mappings to merge identical nodes (connected with `mw:SameAs`) and perform a final deduplication of the edges, resulting in our consolidated CSKG graph. The entire procedure of importing the individual sources and consolidating them into CSKG is implemented with KGTK operations [10], and can be found on our GitHub.[6]

---

[5] `https://github.com/globalwordnet/ili`

[6] `https://github.com/usc-isi-i2/cskg/blob/master/consolidation/create_cskg.sh`

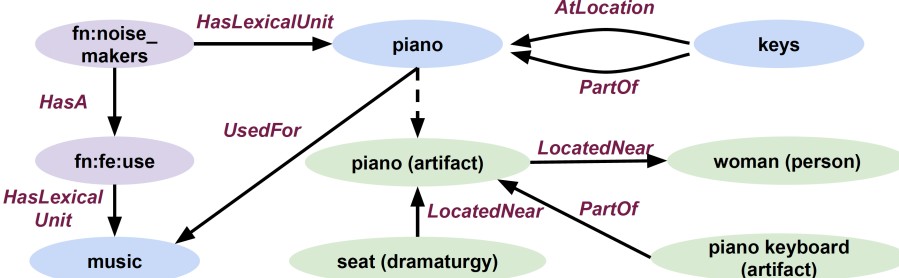

**Fig. 1.** Snippet of CSKG for the example task of section 1. CSKG combines: 1) lexical nodes (piano, keys, music; in blue), 2) synsets like piano (artifact), seat (dramaturgy) (in green), and 3) frames (`fn:noise_makers`) and frame elements (`fn:fe:use`) (in purple). The link between `piano` and `piano (artifact)` is missing, but trivial to infer.

### 3.4   Embeddings

Embeddings provide a convenient entry point to KGs and enable reasoning on both intrinsic and downstream tasks. For instance, many reasoning applications (cf. [18,16]) of ConceptNet leverage their NumberBatch embeddings [29].Motivated by these observations, we aspire to produce high-quality embeddings of the CSKG graph. We experiment with two families of embedding algorithms. On the one hand, we produce variants of popular graph embeddings: TransE [4], DistMult [35], ComplEx [32], and RESCAL [24]. On the other hand, we produce various text (Transformer-based) embeddings based on BERT-large [8]. For BERT, we first create a sentence for each node, based on a template that encompasses its neighborhood, which is then encoded with BERT's sentence transformer model. All embeddings are computed with the KGTK operations `graph-embeddings` and `text-embeddings`. We analyze them in section 4.2.

The CSKG embeddings are publicly available at `http://shorturl.at/pAGX8`.

## 4   Analysis

Figure 1 shows a snippet of CSKG that corresponds to the task in section 1. Following P1, CSKG combines: 1) lexical nodes (piano, keys, music), 2) synsets like piano (artifact), seat (dramaturgy) (in green), and 3) frames (`fn:noise_makers`) and frame elements (`fn:fe:use`). According to P2, we reuse edge types where applicable: for instance, we use ConceptNet's `LocatedNear` relation to formalize Visual Genome's proximity information between a woman and a piano. We leverage external links to WordNet to consolidate synsets across sources (P3). We generate further links (P4) to connect FrameNet frames and frame elements to ConceptNet nodes, and to consolidate the representation of `piano (artifact)` between Wikidata and WordNet. In the remainder of this section, we perform qualitative analysis of CSKG and its embeddings.

**Table 2.** CSKG statistics. Abbreviations: CN=ConceptNet, VG=Visual Genome, WN=WordNet, RG=Roget, WD=Wikidata, FN=FrameNet, AT=ATOMIC. Relation numbers in brackets are before consolidating to ConceptNet.

|  | AT | CN | FN | RG | VG | WD | WN | CSKG* | CSKG |
|---|---|---|---|---|---|---|---|---|---|
| #nodes | 304,909 | 1,787,373 | 15,652 | 71,804 | 11,264 | 91,294 | 71,243 | 2,414,813 | **2,160,968** |
| #edges | 732,723 | 3,423,004 | 29,873 | 1,403,955 | 2,587,623 | 111,276 | 101,771 | 6,349,731 | **6,001,531** |
| #relations | 9 | 47 | 9 (23) | 2 | 3 (42k) | 3 | 15 (45) | 59 | **58** |
| avg degree | 4.81 | 3.83 | 3.82 | 39.1 | 459.45 | 2.44 | 2.86 | 5.26 | **5.55** |
| std degree | 0.07 | 0.02 | 0.13 | 0.34 | 35.81 | 0.02 | 0.05 | 0.02 | **0.03** |

**Table 3.** Nodes with highest centrality score according to PageRank and HITS. Node labels indicated in bold.

| PageRank | HITS hubs | HITS authorities |
|---|---|---|
| /c/en/**chromatic**/a/wn | /c/en/**red** | /c/en/**blue** |
| /c/en/**organic_compound** | /c/en/**yellow** | /c/en/**red** |
| /c/en/**chemical_compound**/n | /c/en/**green** | /c/en/**silver** |
| /c/en/**change**/n/wn/artifact | /c/en/**silver** | /c/en/**green** |
| /c/en/**natural_science**/n/wn/cognition | /c/en/**blue** | /c/en/**gold** |

### 4.1   Statistics

**Basic statistics** of CSKG are shown in Table 2. In total, our mappings produce 251,517 `mw:SameAs` links and 45,659 `fn:HasLexicalUnit` links. After refinement, i.e., removal of the duplicates and merging of the identical nodes, CSKG consists of 2.2 million nodes and 6 million edges. In terms of edges, its largest subgraph is ConceptNet (3.4 million), whereas ATOMIC comes second with 733 thousand edges. These two graphs also contribute the largest number of nodes to CSKG. The three most common relations in CSKG are: `/r/RelatedTo` (1.7 million), `/r/Synonym` (1.2 million), and `/r/Antonym` (401 thousand edges).

   **Connectivity and centrality** The mean degree of CSKG grows by 5.5% (from 5.26 to 5.55) after merging identical nodes. Compared to ConceptNet, its degree is 45% higher, due to its increased number of edges while keeping the number of nodes nearly constant. The best connected subgraphs are Visual Genome and Roget. CSKG's high connectivity is owed largely to these two sources and our mappings, as the other five sources have degrees below that of CSKG. The abnormally large node degrees and variance of Visual Genome are due to its annotation guidelines that dictate all concept-to-concept information to be annotated, and our modeling choice to represent its nodes through their synsets. We report that the in-degree and out-degree distributions of CSKG have Zipfian shapes, a notable difference being that the maximal in degree is nearly double compared to its maximal out degree (11k vs 6.4k). To understand better the central nodes in CSKG, we compute PageRank and HITS metrics. The top-5 results are shown in Table 3. We observe that the node with highest PageRank

**Table 4.** Top-5 most similar nodes for `/c/en/turtle/n/wn/animal` (E1) and `/c/en/happy` (E2) according to TransE and BERT.

| TransE | BERT |
|---|---|
| E1 /c/en/chelonian/n/wn/animal | /c/en/glyptemys/n |
| /c/en/mud_turtle/n/wn/animal | /c/en/pelocomastes/n |
| /c/en/cooter/n/wn/animal | /c/en/staurotypus/n |
| /c/en/common_snapping_turtle/n/wn/animal | /c/en/parahydraspis/n |
| /c/en/sea_turtle/n/wn/animal | /c/en/trachemys/n |
| E2 /c/en/excited | /c/en/bring_happiness |
| /c/en/satisfied | /c/en/new_happiness |
| /c/en/smile_mood | at:like_a_party_is_a_good_way_to_.... |
| /c/en/pleased | /c/en/encouraging_person's_talent |
| /c/en/joyful | at:happy_that_they_went_to_the_party |

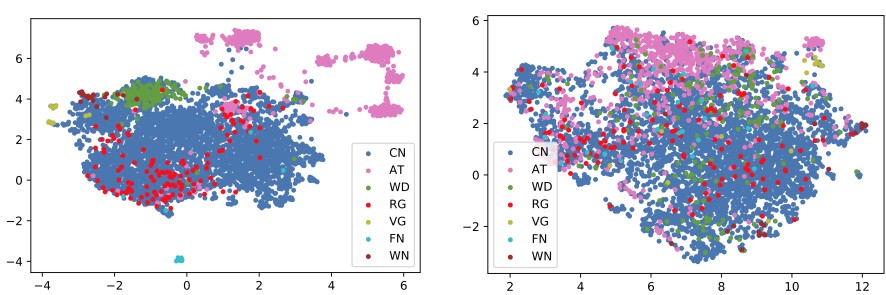

**Fig. 2.** UMAP visualization of 5,000 randomly sampled nodes from CSKG, represented by TransE (left) and BERT (right) embeddings. Colors signify node sources.

has label "chromatic", while all dominant HITS hubs and authorities are colors, revealing that knowledge on colors of real-world object is common in CSKG. PageRank also reveals that knowledge on natural and chemical processes is well-represented in CSKG. Finally, we note that the top-centrality nodes are generally described by multiple subgraphs, e.g., `c/en/natural_science/n/wn/cognition` is found in ConceptNet and WordNet, whereas the color nodes (e.g., `/c/en/red`) are shared between Roget and ConceptNet.

### 4.2   Analysis of the CSKG embeddings

We randomly sample 5,000 nodes from CSKG and visualize their embeddings computed with an algorithm from each family: TransE and BERT. The results are shown in Figure 2. We observe that graph embeddings group nodes from the same source together. This is because graph embeddings tend to focus on the graph structure, and because most links in CSKG are still within sources. We observe that the sources are more intertwined in the case of the BERT embed-

dings, because of the emphasis on lexical over structural similarity. Moreover, in both plots Roget is dispersed around the ConceptNet nodes, which is likely due to its broad coverage of concepts, that maps both structurally and lexically to ConceptNet. At the same time, while ATOMIC overlaps with a subset of ConceptNet [26], the two sources mostly cover different areas of the space.

Table 4 shows the top-5 most similar neighbors for `/c/en/turtle/n/wn/animal` and `/c/en/happy` according to TransE and BERT. We note that while graph embeddings favor nodes that are structurally similar (e.g., `/c/en/turtle/n/wn/animal` and `/c/en/chelonian/n/wn/animal` are both animals in WordNet), text embeddings give much higher importance to lexical similarity of nodes or their neighbors, even when the nodes are disconnected in CSKG (e.g., `/c/en/happy` and `at:happy_that_they_went_to_the_party`). These results are expected considering the approach behind each algorithm.

**Word association with embeddings** To quantify the utility of different embeddings, we evaluate them on the *USF-FAN* [23] benchmark, which contains crowdsourced common sense associations for 5,019 "stimulus" concepts in English. For instance, the associations provided for `day` are: `night`, `light`, `sun`, `time`, `week`, and `break`. The associations are ordered descendingly based on their frequency. With each algorithm, we produce a top-K most similar neighbors list based on the embedding of the stimulus concept. Here, $K$ is the number of associations for a concept, which varies across stimuli. If CSKG has multiple nodes for the stimulus label, we average their embeddings. For the graph embeddings, we use logistic loss function, using a dot comparator, a learning rate of 0.1, and dimension 100. The BERT text embeddings have dimension 1024, which is the native dimension of this language model. As the text embedding models often favor surface form similarity (e.g., associations like `daily` for `day`), we devise variants of this method that excludes associations with Levenshtein similarity higher than a threshold $t$.

We evaluate by comparing the embedding-based list to the benchmark one, through customary ranking metrics, like Mean Average Precision (MAP) and Normalized Discounted Cumulative Gain (NDCG). Our investigations show that TransE is the best-performing algorithm overall, with MAP of 0.207 and NDCG of 0.530. The optimal BERT variant uses threshold of $t = 0.9$, scoring with MAP of 0.209 and NDCG of 0.268. The obtained MAP scores indicate that the embeddings capture relevant signals, yet, a principled solution to USF-FAN requires a more sophisticated embedding search method that can capture various forms of both relatedness and similarity. In the future, we aim to investigate embedding techniques that integrate structural and content information like RDF2Vec [25], and evaluate on popular word similarity datasets like WordSim-353 [9].

## 5  Applications

As the creation of CSKG is largely driven by downstream reasoning needs, we now investigate its relevance for commonsense question answering: 1) we measure

**Table 5.** Number of triples retrieved with ConceptNet and CSKG on different datasets.

|      | train | | | dev | | |
|------|------------|------------|---------|------------|------------|---------|
|      | #Questions | ConceptNet | CSKG | #Questions | ConceptNet | CSKG |
| CSQA | 9,741 | 78,729 | 125,552 | 1,221 | 9,758 | 15,662 |
| SIQA | 33,410 | 126,596 | 266,937 | 1,954 | 7,850 | 16,149 |
| PIQA | 16,113 | 18,549 | 59,684 | 1,838 | 2,170 | 6,840 |
| aNLI | 169,654 | 257,163 | 638,841 | 1,532 | 5,603 | 13,582 |

its ability to contribute novel evidence to support reasoning, and 2) we measure its role in pre-training language models for zero-shot downstream reasoning.

### 5.1   Retrieving evidence from CSKG

We measure the relevance of CSKG for commonsense question answering tasks, by comparing the number of retrieved triples that connect keywords in the question and in the answers. For this purpose, we adapt the lexical grounding in HyKAS [18] to retrieve triples from CSKG instead of its default knowledge source, ConceptNet. We expect that CSKG can provide much more evidence than ConceptNet, both in terms of number of triples and their diversity. We experiment with four commonsense datasets: CommonSense QA (CSQA) [30], Social IQA (SIQA) [27], Physical IQA (PIQA) [3], and abductive NLI (aNLI) [2]. As shown in Table 5, CSKG significantly increases the number of evidence triples that connect terms in questions with terms in answers, in comparison to Concept-Net. We note that the increase is on average 2-3 times, the expected exception being CSQA, which was inferred from ConceptNet.

We inspect a sample of questions to gain insight into whether the additional triples are relevant and could benefit reasoning. For instance, let us consider the CSQA question "Bob the lizard lives in a warm place with lots of water. Where does he probably live?", whose correct answer is "tropical rainforest". In addition to the ConceptNet triple `/c/en/lizard /c/en/AtLocation /c/en/tropical_rainforest`, CSKG provides two additional triples, stating that tropical is an instance of place and that water may have property tropical. The first additional edge stems from our mappings from FrameNet to ConceptNet, whereas the second comes from Visual Genome. We note that, while CSKG increases the coverage with respect to available commonsense knowledge, it is also incomplete: in the above example, useful information such as warm temperatures being typical for tropical rainforests is still absent.

### 5.2   Pre-training language models with CSKG

We have studied the role of various subsets of CSKG for downstream QA reasoning extensively in [19]. Here, CSKG or its subsets were transformed into artificial commonsense question answering tasks. These tasks were then used instead of training data to pre-train language models, like RoBERTa and GPT-2. Such

**Table 6.** Zero-shot evaluation results with different combinations of models and knowledge sources, across five commonsense tasks, as reported in [19]. `CWWV` combines ConceptNet, Wikidata, WordNet, and Visual Genome. `CSKG` is a union of `ATOMIC` and `CWWV`. We report mean accuracy over three runs, with 95% confidence interval.

| Model | KG | aNLI | CSQA | PIQA | SIQA | WG |
|---|---|---|---|---|---|---|
| GPT2-L | ATOMIC | 59.2($\pm$0.3) | 48.0($\pm$0.9) | 67.5($\pm$0.7) | 53.5($\pm$0.4) | 54.7($\pm$0.6) |
| GPT2-L | CWWV | 58.3($\pm$0.4) | 46.2($\pm$1.0) | 68.6($\pm$0.7) | 48.0($\pm$0.7) | 52.8($\pm$0.9) |
| GPT2-L | CSKG | 59.0($\pm$0.5) | 48.6($\pm$1.0) | 68.6($\pm$0.9) | 53.3($\pm$0.5) | 54.1($\pm$0.5) |
| RoBERTa-L | ATOMIC | **70.8($\pm$1.2)** | 64.2($\pm$0.7) | 72.1($\pm$0.5) | 63.1($\pm$1.5) | 59.6($\pm$0.3) |
| RoBERTa-L | CWWV | 70.0($\pm$0.3) | **67.9($\pm$0.8)** | 72.0($\pm$0.7) | 54.8($\pm$1.2) | 59.4($\pm$0.5) |
| RoBERTa-L | CSKG | 70.5($\pm$0.2) | 67.4($\pm$0.8) | **72.4($\pm$0.4)** | **63.2($\pm$0.7)** | **60.9($\pm$0.8)** |
| *Human* | - | 91.4 | 88.9 | 94.9 | 86.9 | 94.1 |

a CSKG-based per-trained language model was then 'frozen' and evaluated in a zero-shot manner across a wide variety of commonsense tasks, ranging from question answering through pronoun resolution and natural language inference.

We select key results from these experiments in Table 6. The results demonstrate that no single knowledge source suffices for all benchmarks and that using CSKG is overall beneficial compared to using its subgraphs, thus directly showing the benefit of commonsense knowledge consolidation. In a follow-up study [11], we further exploit the consolidation in CSKG to pre-train the language models with one dimension (knowledge type) at a time, noting that certain dimensions of knowledge (e.g., temporal knowledge) are much more useful for reasoning than others, like lexical knowledge. In both cases, the kind of knowledge that benefits each task is ultimately conditioned on the alignment between this knowledge and the targeted task, indicating that subsequent work should further investigate how to dynamically align knowledge with the task at hand.

## 6   Discussion

Our analysis in section 4 revealed that the connectivity in CSKG is higher than merely concatenation of the individual sources, due to our mappings across sources and the merge of identical nodes. Its KGTK format allowed us to seamlessly compute and evaluate a series of embeddings, observing that TransE and BERT with additional filtering are the two best-performing and complementary algorithms. The novel evidence brought by CSKG on downstream QA tasks (section 5) is a signal that can be exploited by reasoning systems to enhance their performance and robustness, as shown in [19]. Yet, the quest to a rich, high-coverage CSKG is far from completed. We briefly discuss two key challenges, while broader discussion can be found in [11].

**Node resolution** As large part of CSKG consists of lexical nodes, it suffers from the standard challenges of linguistic ambiguity and variance. For in-

stance, there are 18 nodes in CSKG that have the label 'scene', which includes WordNet or OpenCyc synsets, Wikidata Qnodes, frame elements, and a lexical node. Variance is another challenge, as `/c/en/caffeine`, `/c/en/caffine`, and `/c/en/the_active_ingredient_caffeine` are all separate nodes in ConceptNet (and in CSKG). We are currently investigating techniques for node resolution applicable to the heterogeneity of commonsense knowledge in CSKG.

**Semantic enrichment** We have normalized the edge types across sources to a single, ConceptNet-centric, set of 58 relations. In [11], we classify all CSKG's relations into 13 dimensions, enabling us to consolidate the edge types further. At the same time, some of these relations hide fine-grained distinctions, for example, WebChild [31] defines 19 specific property relations, including temperature, shape, and color, all of which correspond to ConceptNet's `/r/HasProperty`. A novel future direction is to produce hierarchy for each of the relations, and refine existing triples by using a more specific relation (e.g., use the predicate 'temperature' instead of 'property' when the object of the triple is 'cold').

## 7   Conclusions and Future Work

While current commonsense knowledge sources contain complementary knowledge that would be beneficial as a whole for downstream tasks, such usage is prevented by different modeling approaches, foci, and sparsity of available mappings. Optimizing for simplicity, modularity, and utility, we proposed a hyper-relational graph representation that describes many nodes with a few edge types, maximizes the high-quality links across subgraphs, and enables natural language access. We applied this representation approach to consolidate a commonsense knowledge graph (CSKG) from seven very diverse and disjoint sources: a text-based commonsense knowledge graph ConceptNet, a general-purpose taxonomy Wikidata, an image description dataset Visual Genome, a procedural knowledge source ATOMIC, and three lexical sources: WordNet, Roget, and FrameNet. CSKG describes 2.2 million nodes with 6 million statements. Our analysis showed that CSKG is a well-connected graph and more than 'a simple sum of its parts'. Together with CSKG, we also publicly release a series of graph and text embeddings of the CSKG nodes, to facilitate future usage of the graph. Our analysis showed that graph and text embeddings of CSKG have complementary notions of similarity, as the former focus on structural patterns, while the latter on lexical features of the node's label and of its neighborhood. Applying CSKG on downstream commonsense reasoning tasks, like QA, showed an increased recall as well as an advantage when pre-training a language model to reason across datasets in a zero-shot fashion. Key standing challenges for CSKG include semantic consolidation of its nodes and refinement of its property hierarchy. Notebooks for analyzing these resources can be found on our public GitHub page: `https://github.com/usc-isi-i2/cskg/tree/master/ESWC2021`.

## Acknowledgements

This work is sponsored by the DARPA MCS program under Contract No. N660011924033 with the United States Office Of Naval Research, and by the Air Force Research Laboratory under agreement number FA8750-20-2-10002.

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
