# OpenReview forum: "CSKG: The CommonSense Knowledge Graph"
_eswc-conferences.org/ESWC/2021/Conference/Resources_Track — ESWC 2021 Resources_

### Official Review · AnonReviewer1 · 2021-01-10
**Paper 37 review - Updated after rebuttal**

**Rating:** 1
**Confidence:** 2

**Review:**

The paper describes CSKG, the first integrated knowledge graph that connects several common sense knowledge graphs. I think the paper takes an interesting research direction and the work has the potential to benefit the community. However, the paper has several weakness and does not fully articulate the benefits of the reported work.

First and foremost, I don't think the introduction has done a good job motivating this research. I am not sure I fully understand the example here, which does not look like a question and was unclear what the machine is asked to do. Authors should articulate how the different common sense KGs are complementing each other, perhaps with alternative examples, and explain what kind of tasks can benefit from such an integrated KG and how. For example, what is ConceptNet primarily used for research/applications and what about ATOMIC? How exactly could the merge of the two create new usage scenarios?

Second, the related work section does not do a good job explaining the limitations of SoA. The last paragraph mentions a few projects of a similar nature but there is no discussion of what their limitations are. This is problematic because it is difficult for readers to understand how this work differs and what additional values it brings.

Third, during the actual integration, the authors did not explain 1) how these common sense KGs are selected; or 2) how the pairs of KGs were selected for mapping. For 1), there needs to be a strong justification of choosing these seven KGs. If it is not randomly done, then what is the intuition behind? Is it based on maximising the potential benefits of CSKG, but how? For 2), fundamentally the question is how did you choose which KG to map to which others? For example, the authors mentioned 3 pairs of KGs for structured mapping, why so? Why could it not be simply a pair-wise mapping between every two KGs? The choice of such mapping processes needs to follow a systematic approach, which needs to be explained and justified.

Finally, the results do not convey a clear message. I am not sure how to interpret the embedding results on page 11. How do we know if the results are good or bad? What can we compare them against? The results on QA are also not without anomalies. There are a few occasions where CSKG underperformed ATOMIC. What does this mean? Does that indicate problems in the integrated KG? If so, is the problem due to mapping errors, or else?

========Update after rebuttal========
The authors gave a thorough reply addressing most of my questions. I am willing to revise my score in line with other reviewers who are more knowledgeable in this area than me. I would like to ask authors to consider revising their paper to explicitly address some of my questions. At least
- the fact that there is no one-size-fit-all solution when integrating multiple KGs and how this justifies their design
- some reflection on the message we should take from the CSKG results, the fact that it is not beneficial to all cases (your point 6).

**Anonymity:**

No, I would like my review to be deanonymized.

**Strong Points:**

- a good effort on mapping multiple common sense KG
- the number of KGs involved

**Subreviewer:**

I submitted this review.

**Weak Points:**

- related work does not clarify the limitations of SoA
- the methodological design is not clearly justified

---

> ### Author Rebuttal · Authors · 2021-01-28
>
> We thank reviewer 1 (R1) for their extensive review and thought-provoking comments. Clarifications:
> 1) Motivation - Our example on p.1 shows a task of taking a situation (‘woman takes a sit at the piano’) and selecting a reasonable continuation for it (‘she nervously sets her fingers on the keys’). The task is called ‘question answering’ (QA) or ‘inference’ (NLI) [1, 3], despite that there is no formal question, as R1 points out. R1 suggests that “Authors should articulate how the different common sense KGs are complementing each other” - in fact, exactly this is illustrated in par 2 of Section 1, where we show how different KGs contribute complementary evidence to reason over the question. Further comparison is provided in par 3 of Section 1, and Section 2.
> 2) While CSKG might create new usage scenarios, our primary goal is to facilitate joint usage of its heterogeneous graphs, to improve system performance on QA and NLI tasks, as illustrated in the example (point 1) or Sec. 5.2., though sources of CSKG have been used in a broader set of commonsense applications, e.g.: “... [35] combine edges from Visual Genome, WordNet, and ConceptNet in a neural network that produces a scene graph from an image.” (Sec. 2.2).
> 3) ConceptNet and ATOMIC are commonly injected into commonsense reasoning architectures (e.g., HyKAS and KagNet). The sources in CSKG were selected based on their popularity in existing QA/NLI work and the type of knowledge they cover (P2). The mapping development follows our principles P3 and P4 (Sec. 3.1). We fully agree with the reviewer - comprehensive node resolution of these sources is an important future work item (as discussed in Section 6). Yet, we note that the ‘one-method-fits-all’ solution, suggested by the reviewer, would likely not be suitable, given the heterogeneity of the nodes (cf. Section 2.2). A method that would work for the disambiguated nodes in Wikidata and WordNet, cannot be directly applied to the sentence-like nodes in ATOMIC.
> 4) We note that the primary difference CSKG and prior work is exactly as pointed by this reviewer: “The paper describes CSKG, the first integrated knowledge graph that connects several common sense knowledge graphs.” Prior work has integrated pairs or sets of these sources, with the most similar effort being BabelNet - yet, to our knowledge, no prior consolidation aspires to consolidate commonsense knowledge sources into a single graph. Most prior efforts have focused on the lexical sources found in CSKG, and possibly others, without targeting the main commonsense sources, like ConceptNet or ATOMIC. We thank the reviewer for pointing this out - we will enhance the related work section with this information.
> 5) The goal of section 4.2 is to provide an insight into the kind of information captured by representative ‘graph’ and ‘text’ embeddings of CSKG. Note that this is not meant as a rigorous evaluation of the quality of the embeddings, but rather intended to allow researchers to make a suitable choice between the CSKG embeddings according to their needs. We note that a much larger set of 23 embeddings can now be easily computed through the PyKeen [2] library, thanks to its recent support for CSKG. Regarding the magnitude of the results, the MAP of ~0.2 indicates that a principled solution to the association dataset, USF-FAN, requires a more sophisticated embedding search method that can capture various forms of relatedness, instead of only similarity. The discussion of such a method and its design are out of scope of this paper.
> 6) Regarding the results of Section 5.2 - we quote: “ The results demonstrate that no single knowledge source suffices for all benchmarks and that using CSKG is overall beneficial compared to using its subgraphs.” R1’s observation is correct, this is not the case for each instance: “The kind of knowledge that benefits each task is ultimately conditioned on the alignment between this knowledge and the targeted task, indicating that subsequent work should further investigate how to dynamically align knowledge with the task at hand.”. For a broader discussion, we point R1 to: a) the original paper [3], which has an elaborate, hypothesis-driven analysis of these findings; b) our newest work [4], on assessing alignment between knowledge types in CSKG and tasks. Note that both [3] and in [4] are directly enabled by the creation of CSKG as a consolidated resource.
>
> [1] Zellers, R., Bisk, Y., Schwartz, R., & Choi, Y. (2018). Swag: A large-scale adversarial dataset for grounded commonsense inference. arXiv preprint arXiv:1808.05326.
>
> [2] https://github.com/pykeen/pykeen
>
> [3] Ma, K., Ilievski, F., Francis, J., Bisk, J., Nyberg, E., Oltramari, A. Knowledge-driven Data Construction for Zero-shot Evaluation in Commonsense Question Answering.  AAAI’21
>
> [4] Ilievski, F., Oltramari, A., Ma, K., Zhang, B., McGuinness, D. L., & Szekely, P. (2021). Dimensions of commonsense knowledge. arXiv preprint arXiv:2101.04640.

---

### Official Review · AnonReviewer4 · 2021-01-13
**CSKG**

**Rating:** 3
**Confidence:** 5

**Review:**

This work attempts to connect a large number of knowledge graphs together to build an integrated common sense knowledge graph.  Although the techniques to integrate them are not particularly novel, this is a useful resource to the community.  The links are all validated by students before being added to the knowledge graph, and the authors show preliminary evidence and manual evidence that the resource does help in downstream tasks such as question answering by adding triples. The value of such a resource could be useful to a large number of people.

**Anonymity:**

Yes, I would like my review to remain anonymous.

**Strong Points:**

1.  Excellent writing and presentation.
2.  Good attempts to create links that are high quality across the datasets, and have them manually annotated.
3.  Thoughtful design of downstream tasks, such question answering, exploring the embedding space based on different embedding techniques such as BERT and TransE, and outlining their differences.
4.  Accessible resources, data, etc.

**Subreviewer:**

I submitted this review.

**Weak Points:**

This is a very good paper, I see no weaknesses.  I'd like to know if KGTK provides any sort of REST API access to CSKG, and to see a small section on how to use it in a manner that does not require much of a setup effort from users.

---

> ### Author Rebuttal · Authors · 2021-01-28
>
> We genuinely thank reviewer 3 (R3) for their thoughtful review and kind appreciation of our efforts on writing, consolidation method, evaluation, and resource availability. We are happy to see that the reviewer expects our resource to be of value for a large number of people.
>
> We wholeheartedly agree with the reviewer suggestion - in order to bring CSKG closer to its users, it is essential to have a user-friendly API. Our plans are to leverage functionality which is being made available within the KGTK tool. Namely, CSKG will have an intuitive browser, based on SQID [1]. Users will also be able to explore CSKG via text search, as we did for Wikidata [2]. Regarding a programmable API, as the KGTK core operations are based on a uniform Python interface, we intend to provide a Python module that will allow users to seamlessly load CSKG and perform the entire suite of KGTK operations (from graph transformations to analysis and embeddings) as Python functions. Right now, this can already be done through the command line interface of KGTK [3]. Finally, as of January 24th, CSKG is also supported by an independent toolkit, PyKeen [4], which provides an API to compute a wide range of 23 embeddings over this graph.
>
> [1] https://sqid.toolforge.org/#/
>
> [2] https://kgtk.isi.edu/search/
>
> [3] https://github.com/usc-isi-i2/kgtk/tree/dev/examples
>
> [4] https://github.com/pykeen/pykeen

---

### Official Review · AnonReviewer2 · 2021-01-13
**Interesting paper, novel and clear, but lacking in evaluation.**

**Rating:** 1
**Confidence:** 3

**Review:**

The paper presents a "Commonsense KG" based on the merging 7 knowledge sources using 5 integration/mapping methods into a single graph. The paper is easy to read, and the topic is important and relevant to ESWC. The approach/result is novel, however a better description/comparison versus existing work would have been appreciated.
While I enjoyed reading the first part of the paper (Sections 1-3) due to their clarity and the interesting ideas / methods presented, I found the evaluation of the CSKG lacking (see below).

In section 4, the authors evaluate the method by creating embeddings from CSKG using two methods:
a) graph embeddings like TransE, RESCAL, b) "text" embeddings with BERT-large (using a sentence of each node created by its neighborhood).

For evaluating the embeddings the authors use (only) the USF-FAN benchmark (text at end of 4.2). For me, it is unclear what to make of this evaluation results, there is no context, how good are they as compared to other KG embeddings or word embeddings? This subsection needs improvement (see as suggestion next paragraph).

At least according to Table 4 the embeddings are very good at finding similar terms, so why are not traditional intrinsic evaluation datasets for word similarity
(such as WordSim-353, SimLex-999, SemEval-500 (SemEval 2017, Task2), MEN,...) used? ConceptNet Numberbatch has been shown to work very well on these datasets, it would have been interesting to see how well your embeddings perform as compared to traditional word embeddings like Word2Vec or FastText, or compared to CN Numberbatch.
See: https://alt.qcri.org/semeval2017/task2/
Code to easily evaluate an embedding on a number of datasets: https://github.com/kudkudak/word-embeddings-benchmarks
(It is not "required" that CSKG works better on such semantic similarity tasks, as CSKG follows a different approach and goal, but still it would give a good impression of its characteristics).

In Section 5 you start with "As the creation of CSKG is largely driven by downstream reasoning needs", but the evaluation of downstream reasoning with CSKG is very limited. In 5.1. your results are intuitive, but anecdotal evidence based on a single example.
In section 5.2. I am not sure what you mean with "Here, CSKG or its subsets were transformed into artificial commonsense question answering tasks."

In summary, the paper presents a not yet "optimal" solution to the problem of integrating heterogeneous sources of (common-sense, or other) sources of knowledge, but an interesting attempt based on some principles and it is put into action and evaluated.


**Anonymity:**

No, I would like my review to be deanonymized.

**Strong Points:**

(Most strong and weak points were already listed above,  I will not repeat them here).

* Good description of the problem and the  challenges in Section 2.

**Subreviewer:**

I submitted this review.

**Weak Points:**

* The CSKG uses its own format (KGTK) for representing/describing edges. It would have been nice (maybe I missed it) to describe how this format can be mapped to an RDF triple format, esp in the context of a Semantic Web conference.
* Discussion of existing approaches (in Section 2.2) is very brief (too brief) and lacks a comparison to the presented approach in terms of methods used for integration and evaluation strategies and evaluation results of existing work.
* As a PhD student I was told to not use "In [19]" but "In Name et al. [19]" -- but maybe this was just a preference of my then scientific peers.

---

> ### Author Rebuttal · Authors · 2021-01-28
>
> We thank reviewer 2 (R2) for the extensive review and their positive perception of our work. We do our best here to address the key concerns of the reviewer.
> 1) Regarding the related work section (2.2.), most of the consolidation approaches are limited to a pair or a small set of sources, and typically focus on lexical sources. As far as we are aware, CSKG is the first effort to consolidate commonsense sources. Likely the most similar existing resource is BabelNet, which is larger in size and richer in terms of languages and set of sources, yet has a different focus towards lexical and general-purpose sources, rather than commonsense sources (like ConceptNet and ATOMIC). We agree with the reviewer that the paper would benefit from expanding the related work in subsection 2.2 - we will adapt the camera-ready version accordingly.
> 2) We investigate the utility of our embeddings on the USF-FAN dataset, as it has been extensively used in psychology to study cognitive associations made by people. This dataset captures rich associations, based both on similarity (pan - pot), but also on aspects of relatedness, like utility (pan - cook) or location (pan - kitchen), all of which should be captured in optimal embeddings. However, our initial experiments focus on retrieving similar nodes --- it is likely that relatedness queries would require a different usage of the embeddings, which is left for future work. We wholeheartedly agree with the reviewer that established benchmarks, such as WordSim-353, would provide complementary insights into our embeddings, and might be more suitable for an extensive comparison of various CSKG embeddings - we will indicate this in the final version of the paper.
> 3) The goal of section 4 is to provide an insight into the kind of information captured by some representative ‘graph’ and ‘text’ embeddings, to enable researchers to make an informed choice between the two. Note that we do not aspire to provide an extensive study of the quality of various CSKG embeddings. Such an extensive study would be facilitated with the most recent inclusion of CSKG in the PyKeen package [1], which allows computing 23 different embeddings, including RotatE and HolE, over CSKG. However, such a study is out of scope of our paper, we will clarify this in the camera-ready version.
> 4) Quantitative evaluation of using CSKG for downstream reasoning is provided in section 5.2. The method of enhancing commonsense question answering baselines with CSKG knowledge is elaborated in the referenced paper [2]. The key idea of this method is to use the knowledge in CSKG to create multiple-choise questions, that are in turn used to pretrain language models. The enhanced language models are then able to answer questions on unseen datasets better, thus showing a clear impact of using CSKG. The experiments show that using CSKG for this purpose is overall superior than using its subsets (ConceptNet or ATOMIC), directly showing the benefit of commonsense knowledge consolidation.
> 5) Regarding mapping CSKG to RDF, this can be done by using the `kgtk generate-wikidata-triples` command [3]. We will also publish CSKG in a SPARQL endpoint as we have done with other graphs that we generated using KGTK, like [4].
>
> [1] https://github.com/pykeen/pykeen
>
> [2] Knowledge-driven Data Construction for Zero-shot Evaluation in Commonsense Question Answering. Kaixin Ma, Filip Ilievski, Jonathan Francis, Yonatan Bisk, Eric Nyberg, Alessandro Oltramari. AAAI’21
>
> [3] https://kgtk.readthedocs.io/en/latest/export/generate_wikidata_triples/
>
> [4] https://dsbox02.isi.edu:8888/wikidataos/

---

### Official Review · AnonReviewer5 · 2021-01-14
**Nice contribution**

**Rating:** 1
**Confidence:** 3

**Review:**

This paper faces the problem of integrating different sources of background knowledge and show their potential in intrinsic and downstream applications. The topic is very interesting; indeed, while there are several sources of background knowledge covering different aspects (authors point out several of them. from FrameNet to ATOMIC) providing a consistent view over all of them is clearly a challenging and very useful task. Authors base their technique on 5 principles meant to address challenges like modeling diversities and sparse overlap. The starting point of the technique is a set of existing links between sources of background knowledge mainly using WordNet synsets as hubs.

Authors describe the engineering beyond their proposal and discuss in Section 4 some statistics and embeddings (the visualization of a sample of 5000 nodes) and in Section 5 a common-sense question answering task.


-- WordNet-Wikidata: providing more information about the 17 students taking part in the task would be useful.


**Anonymity:**

Yes, I would like my review to remain anonymous.

**Strong Points:**

-Paper clear and well-written

-Both the code and the datasets are made available. Moreover, there is a very clear and comprehensive "guide" to the usage.

-The analysis of the resulting graph (Section 4) and potential applications (Section 5) is a plus.

**Subreviewer:**

I submitted this review.

**Weak Points:**

- From a research perspective, this paper does not offer too much. It is mostly engineering work and most of the paper is devoted to describing practical aspects

-It seems that the Babelnet license (https://babelnet.org/license) does allow for extensions

---

> ### Author Rebuttal · Authors · 2021-01-28
>
> We thank reviewer 5 (R5) for their insightful review and words of appreciation for our work. We would like to clarify the matters pointed by the reviewer:
>
> 1) The paper shows that there is vast heterogeneity of existing sources of commonsense knowledge, which prevents their joint usage on downstream tasks (like QA). Besides our effort on bridging this gap, our paper reveals a rich set of integration challenges, primarily on node resolution and semantic enrichment (cf. Section 6), but also others which we did not include due to space constraints, such as matters of data quality and relevance (e.g., part of ConceptNet describes cultural or science-specific knowledge). Our method addresses the listed challenges to some extent, and we study abstraction of commonsense knowledge types in a follow-up journal submission [1] --- however, optimal common sense consolidation is not a solved task yet.
>
> 2) The main page of BabelNet [2] states that BabelNet is “ResponsiveVoice-NonCommercial licensed under CC BY-NC-ND”. Such a license does allow sharing, but not adaptations of the content (the latter is needed for integrating parts of it in CSKG). Yet, the license site of BabelNet pointed by R5 [3] indicates in ‘human-readable’ terms that BabelNet can be both shared and remixed/adapted. It seems like these two statements are contradictory - we will contact the authors to understand the license better, and if allowed by the license, we would be interested to include some of the mappings offered by BabelNet.
>
> [1] Ilievski, F., Oltramari, A., Ma, K., Zhang, B., McGuinness, D. L., & Szekely, P. (2021). Dimensions of commonsense knowledge. arXiv preprint arXiv:2101.04640.
>
> [2] https://babelnet.org/about
>
> [3] https://babelnet.org/license

---

### Official Review · AnonReviewer3 · 2021-01-15
**Very promising resource with solid methodological background**

**Rating:** 2
**Confidence:** 4

**Review:**

The paper describes CSKG: The CommonSense Knowledge Graph, a Knowledge Graph that integrates existing resources that represent commonsense knowledge of different nature.
CSKG is a very promising resource and I am definitely willing to try and use it.
The creation of the CSKG is well motivated and executed with a sound methodology.  I do not object to the approach proposed to the authors. I liked the publication of the graph as well as of the embeddings computed with different methods.
The paper is also well written (despite I suggest some changes), convincing, and easy to read.  I liked very much the introduction which provides a very good map that surveys different useful resources.
I only have a couple of remarks to consider for the final version of the paper if it is accepted.
1.	I think that the comparison with BabelNet should be extended, because, IMHO, BabelNet is the KB most similar to CSKG. Even if BabelNet is not open, as a reader I would like to better understand how the two resources differ one from the another.
2.	I suggest inserting one example of nodes that shows how the different components are integrated. I understand that space is limited and that technical details were important for evaluating the paper. But if accepted, I would rather prefer to have an intuitive presentation of the resulting KG (with an example, e.g., of piano) in this paper and have a link to a more complete description of the approach. On the other side, some remarks about the specific sources are not easy to ponder if one has limited knowledge of the described resource (although most of them are known in our community).  As a consequence, I would favor having an extended paper with these details (and maybe some more expanded examples), e.g., in an open repository, and link to that paper, while concluding somehow the running example showing how the CSKG supports the input scenario.

### Additional comments

Do you publish the embeddings computed with all the models reported in the paper? This is interesting. In general, however, KG embeddings like TransE, DISTMulti, etc., are optimized over link prediction tasks. Models inspired to distributional semantics like RDF2VEC could instead provide embeddings that better capture relatedness, which may be more useful in some contexts (for example, KG embeddings optimized over link prediction tasks usually are not very good at resolving analogies). Do you plan to include such orthogonal embeddings model in the future?

### After rebuttal

I have read other reviews and authors' replies. I prefer to stick to my opinion: I think this is a nice resource paper (somehow I feel like stressing this qualification), which describes an equivalently interesting resource that has been constructed with a thoughtful methodology. Will the resource be used? Apparently, there's an AAAI paper already using it. I think that more uses will come in the future. Does it partially overlap with BabelNet? It may, partially, overlap, but the authors have discussed the differences in the rebuttal and my impression is that the two resources are helpful in different downstream tasks.


**Anonymity:**

Yes, I would like my review to remain anonymous.

**Strong Points:**

* Many different aspects of commonsense in one place
* The resource itself as well as its embeddings are openly available
* Clear potential impact on future work.


**Subreviewer:**

I submitted this review.

**Weak Points:**

* A more detailed comparison with BabelNet is missing
* An example of how information is structured in the final graph is missing.

---

> ### Author Rebuttal · Authors · 2021-01-28
>
> We thank reviewer 3 (R3) for their appreciation of our work and the mindful review. We are grateful for the suggestions, and we will:
> 1) Expand the related work review to include a more informative comparison with BabelNet - BabelNet contains more sources than CSKG, covers a wide range of 284 languages, and primarily focuses on lexical and general-purpose resources, like WordNet, VerbNet, and Wiktionary. While CSKG shares the goal of integrating valuable sources for downstream reasoning, and some of these sources (e.g., WordNet) overlap with BabelNet, our ambition is to support commonsense reasoning applications. For this reason, most of CSKG is based on commonsense knowledge graphs, like ConceptNet and ATOMIC, or even visual sources, like Visual Genome, none of which are found in BabelNet.
> 2) Include a figure to illustrate a snippet of the CSKG knowledge graph that corresponds to the piano use case - The figure will show how we model and harmonize the background knowledge that we discuss in Section 1 (e.g., pianos have keys, keys can be black, etc.), and it will indicate the specific mappings used to consolidate each node.
> 3) CSKG’s subgraphs like ConceptNet and ATOMIC contain textual nodes. Hence, as R3 points out, it is intuitive that, besides the graph structure (optimized through tasks like link prediction), we need to consider the content of the node label (which could be captured through distributional semantics embeddings). For this reason, we considered both ‘graph’ embeddings (like TransE) and ‘text’ embeddings (based on BERT). We note that, as of January 24th, CSKG is included in the PyKeen package[1], which allows computing 23 different embeddings, including RotatE and HolE, over CSKG. Yet, using models that natively combine structure and content, like RDF2Vec, is a valuable suggestion and we will include this in the future work section.
> As R3 suggests, to make space for the additions, we will move some of the details on the individual source modeling/extraction (section 3.3.1) into a Wiki on CSKG’s GitHub page. We note that we prepared a broader survey of commonsense sources with further information on the their modeling and knowledge types in [2].
>
> [1] Ilievski, F., Oltramari, A., Ma, K., Zhang, B., McGuinness, D. L., & Szekely, P. (2021). Dimensions of commonsense knowledge. arXiv preprint arXiv:2101.04640.
>
> [2] https://github.com/pykeen/pykeen

---

> > ### Comment · AnonReviewer3 · 2021-02-01
> > **Staying with my comments**
> >
> > I have read other reviews and authors' replies. I prefer to stick to my opinion: I think this is a nice resource paper (somehow I feel like stressing this qualification), which describes an equivalently interesting resource that has been constructed with a thoughtful methodology. Will the resource be used? Apparently, there's an AAAI paper already using it. I think that more uses will come in the future. Does it partially overlap with BabelNet? It may, partially, overlap, but the authors have discussed the differences in the rebuttal and my impression is that the two resources are helpful in different downstream tasks.

---

### Decision · Program_Chairs · 2021-02-23

**Decision:**

Accept

**Comment:**

The resource is considered original, impactful and adequately available, and the text mostly well written. The evaluation is present, though somewhat limited. Some improvements should be made in the final version of the paper regarding the comparison with related research and justification of some design decisions.